# Optimization Extraction and Antioxidant Activity of Crude Polysaccharide from Chestnut Mushroom (*Agrocybe aegerita*) by Accelerated Solvent Extraction Combined with Response Surface Methodology (ASE-RSM)

**DOI:** 10.3390/molecules27082380

**Published:** 2022-04-07

**Authors:** Bin Du, Fei Peng, Ying Xie, Huiying Wang, Jinhui Wu, Chang Liu, Yuedong Yang

**Affiliations:** 1Engineering Research Center of Chestnut Industry Technology, Ministry of Education, Hebei Normal University of Science and Technology, Qinhuangdao 066004, China; bindufood@aliyun.com (B.D.); flyer5212528@163.com (F.P.); flyfire1103@163.com (Y.X.); wanghuiyinghyw@163.com (H.W.); wujinhuichemistry@163.com (J.W.); lch2647@hevttc.edu.cn (C.L.); 2Hebei Collaborative Innovation Center of Chestnut Industry, Hebei Normal University of Science and Technology, Qinhuangdao 066004, China; 3Hebei Key Laboratory of Natural Products Activity Components and Function, Hebei Normal University of Science and Technology, Qinhuangdao 066004, China

**Keywords:** chestnut mushroom, accelerated solvent extraction, polysaccharide, response surface methodology, antioxidant activity

## Abstract

The present work is conducted to investigate the optimal extraction technology of polysaccharide from chestnut mushroom (*Agrocybe aegerita*) using a new method based on accelerated solvent extraction combined with response surface methodology (ASE-RSM). The conventional reflux extraction (CRE) method and ultrasonic-assisted extraction (UAE) method were also carried out. Additionally, the in vitro antioxidant activities, including ABTS and DPPH assay, were evaluated. The RSM method, based on a three level and three variable Box–Behnken design (BBD), was developed to obtain the optimal combination of extraction conditions. In brief, the polysaccharide was optimally extracted with water as extraction solvent, extraction temperature of 71 °C, extraction time of 6.5 min, number of cycles of 3, and extraction pressure of 10 MPa. The 3D response surface plot and the contour plot derived from the mathematical models were applied to determine the optimal conditions. Under the above conditions, the experimental value of polysaccharide yield was 19.77 ± 0.12%, which is in close agreement with the value (19.81%) predicted by the model. These findings demonstrate that ASE-RSM produce much higher polysaccharide and consumed environmentally friendly extraction and solvent systems, have less extraction discrimination and shorter time and provide scientific basis for industrialization of polysaccharide extraction. Moreover, it was proved that the polysaccharide had the potential ability to scavenge ABTS and DPPH.

## 1. Introduction

Mushrooms have been valuable as edible and medical provisions for humankind. Mushrooms have nutritional value and unique taste and specific flavor [1]. It is attributed not only to the quality of their taste but also to their biological activities, including antidiabetic, antioxidative, antitumor, and anti-inflammatory effects [2]. Chestnut mushroom (*Agrocybe aegerita*) is a white rot basidiomycete [3]. It is found in Asia, North America and Europe [4]. Mushrooms accumulate a number of compounds: carbohydrates, fibers, vitamins, minerals, proteins, fats, and different secondary metabolites [5]. These mycochemicals in mushroom play important roles in preventing and treating oxidative stress, cell damage, and various chronic diseases [6]. In recent years, growing attention has been paid to mushroom polysaccharide due to its biological properties, such as anti-inflammatory [7,8,9], antioxidant [10,11], antitumor activity [12], immunomodulatory effects [13,14], and prebiotic function [15].

In terms of polysaccharide extraction, a lot of methods based on different principles have been developed. For instance, hot water extraction, hot alkaline extraction, ultrasound-assisted extraction, microwave-assisted extraction, superheated water extraction, and enzymatic method extraction have been applied to extract *β*-glucan from barley [16]. These traditional extraction methods had disadvantages of long times and high temperatures, low efficiency, and possible polysaccharides degradation, resulting in large consumptions of energy and time [17]. The accelerated solvent extraction (ASE) technique is a relatively new automatic extraction technique, which uses high pressure. This technique allows the user to carry out extractions at temperature above the boiling temperature of the solvent used [18]. This is a green process, and the method has the advantages of simplicity in procedures, high product quality, high efficiency, and low cost. This is a short-time process without chemical solvent; treatments that may degrade the polysaccharide and affect the activity of polysaccharide. Owing to a fully automated and reliable extraction technique with many advantages [19], extensive application of ASE has been reported in food and pharmaceutical fields [20,21,22]. Therefore, special attention should thus be given to optimize ASE of polysaccharide from chestnut mushroom and their industrial exploitation.

Response surface methodology (RSM) is an effective statistical technique to evaluate the effects of influential factors on one or more response variables for optimizing complex extraction processes [23,24]. The accelerated solvent extraction combined with response surface methodology (ASE-RSM) has been applied for the extraction of natural products [25,26].

In this work, the ASE-RSM methods were used to optimize the operational parameters (extraction temperature, extraction time, number of cycles) to obtain the maximum yield of polysaccharide from chestnut mushroom. RSM was designed through employing a Box–Behnken design (BBD) to systematically analyze the effects of extraction parameters on the yields of polysaccharide from chestnut mushroom and their interactions. This research will be used as a basis for further studies for biological research, as well as in the scale-up of the mushroom polysaccharide extraction process for industrial purposes.

## 2. Results and Discussion

### 2.1. Comparison of Different Extraction Methods

The yields of crude polysaccharide using different extraction methods are shown in Table 1. The highest yield of crude polysaccharide (19%) was obtained under ASE, followed by conventional reflux extraction (CRE) method (3.2%), and ultrasonic-assisted extraction (UAE) method (1.3%). Compared with CRE (2 h) and UAE (1 h), ASE method consumed only 6.5 min. In addition, ASE produced much higher polysaccharide and spent less time. The CRE method has distinct draw backs, such as labor-intensive operations, time-consuming, and extended concentration steps, which can lead to the degradation of polysaccharides [21].

### 2.2. Effect of Extraction Temperature on Yield of Crude Polysaccharide

The extraction process was performed at different temperature conditions (40, 50, 60, 70, and 80 °C) while other extraction variables were set as follow: extraction time of 6 min, extraction pressure of 10 MPa, and number of cycles of 2. As shown in Figure 1a, the crude polysaccharide yield reached a maximum at the temperature of 70 °C and then started to reduce. This tendency was consistent with the study of other authors in extracting polysaccharides [21,27]. This revealed that temperature enhanced the crude polysaccharide extraction from chestnut mushroom into the water to a certain level followed by their possible loss, due to decomposition at a higher temperature.

### 2.3. Effect of Extraction Time on Yield of Crude Polysaccharide

The effect of extraction time on yield of crude polysaccharide is shown in Figure 1b. The extraction times were set at 2, 4, 6, 8, and 10 min, other experimental conditions were as follows: extraction temperature of 70 °C, extraction pressure of 10 MPa, and number of cycles of 2. The results showed that the extraction yield improved over time until 6 min and then began to decline, and the maximum extraction yield was 18.11% at 6 min. The decreases of crude polysaccharide yield might be owing to longer extraction time inducing the degradation of carbohydrate [28].

### 2.4. Effect of Number of Cycles on Yield of Crude Polysaccharide

The number of cycles was an important factor that would influence the extraction yield. To study the effect of different numbers of cycles on the crude polysaccharide yield, the extraction process was carried out using different numbers of cycles (1, 2, 3, and 4). Other extraction variables were set as follows: extraction time of 6 min, extraction pressure of 10 MPa, and extraction temperature of 70 °C. The crude polysaccharide yield achieved a maximum when the number of cycles was two (Figure 1c), while when more than two cycles, the results observed more cycles and less yield. The possible reason is that crude polysaccharide had highest content with two cycles. It does not work for a greater number of cycles.

### 2.5. Formatting of Mathematical Co-odel Building and Statistical Analysis

There was a total of 17 runs for optimizing the three individual parameters in the current Box–Behnken design. The current design was applied to the production of crude polysaccharide by ASE. Table 2 shows the process variables and experimental data. The *F*-value of the lack of fit divided by the pure error (784.11) is higher. Consequently, there is a lack of fit. The lack of fit arises from the low variability in the central points and low pure error. The results of the analysis of variance (ANOVA), goodness-of-fit, and the adequacy of the models were summarized. By applying multiple regression analyses on the experimental data, the response variable and the test variables were associated with the following second-order polynomial Equation (1):*Y* = 19.21 + 0.12*A* + 0.89*B* + 2.175*C* − 0.69*AB* − 0.74*AC* + 0.45*BC* − 3.69*A*^2^ − 1.38*B*^2^ − 2.30*C*^2^(1)

The fit statistics of extraction yield (*Y*) for the selected quadratic predictive model are shown in Table 3. For the model fitted, the coefficient of determination (*R*^2^) was 0.9319, indicating that only 6.81% of the total variations were not explained by the model. *F*-value for the lack of fit was insignificant (*p* > 0.05), thereby confirming the validity of the model. The value of the adjusted determination coefficient (adjusted *R*^2^ = 0.9319) was almost 1, indicating a high degree of correlation between the observed and predicted values. At the same time, a low coefficient of the variation (CV) value 7.82 clearly indicated a very high degree of precision and a good deal of reliability of the experimental values. The model *p*-value (Prob > *F*) was very low, which implied that the model was highly significant.

The model was found to be adequate for prediction within the range of experimental variables. The regression coefficient values of Equation (1) are listed in Table 4. The *p*-value was applied as a tool to assess the significance of each coefficient, which in turn might indicate the pattern of the interaction between the variables. The smaller the value of *p* was, the more significant the corresponding coefficient was [29,30]. Table 3 shows that the linear coefficients (*C*) and quadratic term coefficients (*A*^2^ and *C*^2^) were significant, with very small *p*-value (*p* < 0.05). The other term coefficients were not significant (*p* > 0.05). Therefore, *C*, *A*^2^, and *C*^2^ were important factors in the extraction process of the crude polysaccharide.

### 2.6. Optimization of Procedure

The graphical representations of the regression Equation (1), called the response surfaces and the contour plots were analyzed using Design-Expert software, and the results of extraction yield of crude polysaccharide affected by extraction temperature, extraction time, and number of cycles are presented in Figure 2. The relationship between independent and dependent variables is indicated in 3D representation of the response surface plots generated by the model of extraction yield; two variables were depicted in one 3D surface plot while the other variables remained at zero level. As shown in Figure 2a, when the number of cycles (*C*) was fixed at zero level, extraction temperature (*A*) and extraction time (*B*) showed reciprocal interaction on extraction yield. It was shown that extraction temperature and extraction time demonstrated quadratic effects on the extraction yield. When extraction temperature (*A*) remained at a lower level, the yield increased at first, then decreased with the increase of time. Figure 2b shows the extraction temperature (*A*) and number of cycles (*C*) demonstrating quadratic effects on the yield of crude polysaccharide when extraction time (*B*) was fixed at level zero. When extraction temperature (*A*) remained at lower level, the yield increased at first, then decreased with the increases of number of cycles (*C*). Figure 2c presents the 3D response surface plots at varying extraction times (*B*) and number of cycles (*C*) at fixed extraction temperature (*A*) (0 level). It shows that the extraction yield was influenced significantly by extraction time and number of cycles. When extraction time (*B*) remained at lower level, the yield increased significantly with the increase at first, then reduced with the number of cycles (*C*).

The optimal extraction conditions of polysaccharide from chestnut mushroom were extraction temperature at 70.84 °C, extraction time of 6.34 min, and number of cycles of 2.43. Among the three extraction parameters studied, the number of cycles was the most significant factor affecting the extraction yield of crude polysaccharide, followed by extraction temperature and extraction time according to the regression coefficients’ significance of the quadratic polynomial model (Table 2) and gradient of slope in the 3D response surface plots (Figure 2).

### 2.7. Validation of Model

The suitability of the model equation for predicting the optimum response values was tested using the optimal parameters with minor modifications. The experimental yield and maximum predicted yield of polysaccharide from chestnut mushroom were compared. Additional experiments using the predicted optimum parameters for polysaccharide extraction were as follows: extraction temperature of 70.84 °C, extraction time of 6.34 min, and number of cycles of 2.43, and the model predicted a maximum response of 19.81%. In order to ensure the predicted result was not biased toward the actual value, experiment rechecking was performed using these modified parameters: extraction temperature of 71 °C, extraction time of 6.5 min, and number of cycles of 3. A mean value of 19.77 ± 0.12% (*n* = 3) obtained from real experiments, demonstrated the validation of the RSM model. The good correlation among these results confirmed that the response model was adequate for reflecting the expected optimization. The results of the analyses indicated that the experimental values were in good agreement with the predicted one, and also suggested that the model of Equation (1) was accurate and satisfactory. Moreover, the extraction yields of polysaccharides from medicinal mushroom *Ganoderma lucidum* reached 0.63% under the optimal ultrasonic-assisted extraction conditions [31]. In another study, the yield of beta-glucan from hull-less barley was 16.39 ± 0.3% [21].

### 2.8. The IR Spectrum of Crude Polysaccharides

The IR spectrum of isolated exopolysaccharide was shown in supplemental Figure 3. FT-IR spectra collected confirm the composition of each of the extracted carbohydrates or mixtures. A strong band at 3390 cm^−^^1^ was assigned to the hydroxyl stretching vibration of the polysaccharide. The high absorbency range of 1085 cm^−^^1^ was the characteristic absorption peak of polysaccharide. A strong stretching vibration at around 1700 cm^−1^ (C=O) implied the presence of carboxyl groups.

### 2.9. Molecular Weight (MW) of Crude Polysaccharide

The MW of crude polysaccharide from chestnut mushroom was 2300 KDa. Another report indicated that the average molar mass weight (Mw) of *Pleurotus eryngii* polysaccharide was 8.372 × 10^6^ g/mol [32]. It can be speculated that the different MW obtained might depend on the natures of the starting material, extraction temperature, extraction time, and fractionation methodology.

### 2.10. DPPH Free Radical Scavenging Capacity of Polysaccharide

As a rapid and simple measure of antioxidant activity, the DPPH radical scavenging capacity has been widely used. The DPPH radical assay results showed that the polysaccharide exhibited in vitro antioxidant activity (Figure 4). In a certain concentration range, the scavenging ability of polysaccharide to DPPH was positively correlated with the concentration of polysaccharide. When the concentration of polysaccharides reaches 0.16 mg/mL, the scavenging rate of DPPH is 26.03%. The scavenging effect of polysaccharide and ascorbic acid on the DPPH radical decreased in the order of ascorbic acid > polysaccharide. Moreover, the different results might be related to the different nature of dietary fiber, such as mushroom and other materials. There were some other studies presenting the different phenomena apart from this work. In one study, He et al. [32] investigated the antioxidant activity of polysaccharide extract from spent mushroom substrate of *Pleurotus eryngii*. The polysaccharide exhibited strong antioxidant activities in a dose dependent manner, using different in vitro models, such as ABTS, superoxide, hydroxyl, and DPPH. It was then speculated that the different phenomena depended on different treatments and experimental instruments.

### 2.11. ABTS Free Radical Scavenging Capacity of Polysaccharide

ABTS method determines the antioxidant activity of hydrogen-donating antioxidants and of chain-breaking antioxidants. The experimental results showed that the scavenging ability of polysaccharide to ABTS was positively correlated with the concentration of polysaccharide in a certain concentration range. The antioxidant capacities of polysaccharide and ascorbic acid using the ABTS assay are shown in Figure 5. In the present study, the ascorbic acid and polysaccharide showed notable ABTS cation radical scavenging activity. In similar research, acetylated polysaccharide from mushroom *Inonotus obliquus* resulted in higher antioxidant abilities [33].

## 3. Materials and Methods

### 3.1. Mushroom Sample

Chestnut mushroom was purchased from Qianxi County, Hebei province. The samples were dried at 50 °C using a vacuum oven (DZF-6000, Shanghai Binglin Ltd., Company, Shanghai, China) for 12 h and were ground into powder by a laboratory mill (JSP-200, Yongkang Jinsui Machinery Manufacturing Factory, Jinhua, China) and passed through a 40-mesh sieve for crude polysaccharide extraction.

### 3.2. Chemical and Reagents

α-Amylase and protease were purchased from Beijing Boao Biological Co., Ltd. (Beijing, China). All other chemicals were analytical grade.

### 3.3. Accelerated Solvent Extraction

Polysaccharide was extracted from chestnut mushroom using ASE method by an APLE-3000 system (Beijing Titan Instruments Co., Ltd., Beijing, China). Nitrogen was obtained from a gas cylinder to assist the pneumatic system and to purge the extraction cells. Dried ground chestnut mushroom (10.0 g) was mixed with celite and submitted to extraction. The mixture packed into a 100 mL pressure-resistant stainless steel extraction cell was extracted with distilled water at the extraction pressure of 10 MPa. For preliminary studies, single-factor experiments were performed by varying the temperature (40–80 °C), number of cycles (1–4 cycles), and extraction time (2–10 min). The extract was used for further study.

### 3.4. Conventional Reflux Extraction (CRE)

A 10.0 g sample was extracted with 150 mL of distilled water under reflux at 100 °C for 2 h. Then the mixture was filtered through Whatman No.1 filter paper (Whatman-Xinhua Filter Papers Co., Hangzhou, China). The filtrate was used for further study.

### 3.5. Ultrasound-Assisted Extraction (UAE)

The ultrasound-assisted extraction was performed in an ultrasonic cleaning bath (KQ3200B, 40.0 kHz, 150 W, Kunshan Ultrasonic Instrument Co., Ltd., Suzhou, China) with a usable capacity of 2.5 L (the internal dimensions: 30.0 × 15.0 × 15.0 cm). An in-water pipe was added to the opposite out-water pipe in the bath, and the flux ratio between in-water and out-water was regulated to keep the solution temperature stable in the test. Samples were placed into a conical flask (150 mL), made up to the required volume with distilled water, and sonicated at the required temperature for different times. Then the mixture was filtered through Whatman No.1 filter paper (Whatman-Xinhua Filter Papers Co., Hangzhou, China). The filtrate was used for further study.

### 3.6. Preparation of Crude Polysaccharide

The crude polysaccharide was prepared by referring to the method of our published paper [21] with necessary modification. The percentage of crude polysaccharide yield (%) was calculated as follow:

Crude polysaccharide yield (%) = (Weight of crude polysaccharide)/(Weight of chestnut mushroom powder) × 100%

### 3.7. Experimental Design

After determining the preliminary range of ASE variables through a single test, a three-variable-three-level BBD [34] was applied to optimize the extraction condition in order to obtain the high yield of crude polysaccharide from chestnut mushroom. The three independent variables were extraction temperature (°C, *A*), extraction time (min, *B*), and number of cycles (*C*), and each variable was set at three levels. A total of 17 experiments were designed according to BBD. Each experiment was performed in triplicate and the average yield of crude polysaccharide (%) was taken as the response, *Y*.

Regression analysis was performed for the experimental data and was fitted into the empirical second order polynomial model, as shown in the Equation (2):(2)$$Y=A_{0}+\overset{3}{\underset{i=1}{\sum }}A_{i}X_{i}+\overset{3}{\underset{i=1}{\sum }}A_{ii}X_{i}^{2}+\overset{2}{\underset{i=1}{\sum }}\overset{3}{\underset{j=i+1}{\sum }}A_{ij}X_{ij}$$
where *Y* was the dependent variable; *A*_0_ was constant; and *A_i_*, *A_ii_*, and *A_ij_* were coefficients estimated by the model. *X_i_* and *X_j_* were levels of the independent variables.

### 3.8. Measurement of Infrared (IR) Spectrum of Polysaccharide

The IR spectrum of the exopolysaccharide was determined using a TENSOR 27 FT-IR spectrophotometer (Bruker Corporation, Karlsruhe, Germany). The sample was ground with spectroscopic grade KBr powder and then pressed into 1 mm pellets for FT-IR determination in the frequency range of 4000–400 cm^−1^.

### 3.9. Measurement of Molecular Weight (MW)

Crude polysaccharide was characterized for molecular weight using an Agilent 1100 series HPLC system (Agilent Technologies, Palo Alto, CA, USA) equipped with a TOSOHTSK-GELG 3000 SWXL column (7.8 mm × 30 cm, 10 µm; TOSOH Corp., Tokyo, Japan) and a refractive index detector. A sample of 20 µL was injected into the system by maintaining a flow rate of 0.5 mL min^−1^ and column temperature of 35 °C. Separation was carried out using 0.05 mol L^−1^ phosphate buffer (pH 6.7) containing 0.05% NaN_3_ as mobile phase. The average molecular weight was calculated by the calibration curve obtained using various standard dextrans (738, 5800, 11,220, 21,370, 41,800, 110,000, 118,600, 318,000, and 813,500).

### 3.10. DPPH Free Radical Scavenging Capacity Assay

DPPH radical scavenging capacity of samples was evaluated according to the method of Luo [35] with slight modifications. Two milliliters of sample extract (1 mg mL^−1^) and 2.0 mL of 0.1 mmol/L DPPH in ethanol were briefly mixed in a test tube. Before standing in darkness for 30 min at 37 °C, the mixture solution was mixed for 1 min with a vortex. Ascorbic acid was used as the positive control and ethanol was used as the blank. The absorbance was measured at 517 nm with a visible light spectrophotometer. The experiment was carried out in triplicate. Radical scavenging activity was calculated using the following formula:%DPPH inhibition = (1 − (A_1_ − A_2_)/A_0_)× 100
where 

A_0_ = absorbance of 2 mL DPPH-ethanol + 2 mL ethanol;A_1_ = absorbance of 2 mL DPPH-ethanol + 2 mL sample;A_2_ = absorbance of 2 mL ethanol + 2 mL sample.

### 3.11. ABTS Free Radical Scavenging Capacity Assay

ABTS radical cation (ABTS) was produced by reacting ABTS solution (7.4 mM) with 2.6 mM K_2_S_2_O_8_ and allowing the mixture to stand in the dark at room temperature for 12–16 h before use. For the study, the ABTS solution was diluted in deionized water or ethanol to an absorbance of 0.7 (±0.02) at 734 nm. An appropriate solvent blank reading was taken. After the addition of 100 μL of sample solutions (1 mg mL^−1^) to 3 mL of ABTS solution, the absorbance reading was taken at 30 °C, 10 min after initial mixing. All solution was used on the day of preparation, and all determinations were carried out in triplicate [36]. The percentage of inhibition of ABTS was calculated using the following formula:%ABTS inhibition = (1 − (A_1_ − A_2_)/A_0_) × 100
where: A_0_ = absorbance of ABTS + dilute solution + water;A_1_ = absorbance of ABTS + dilute solution + sample;A_2_ = absorbance of PBS + sample.

### 3.12. Statistical Analysis

Design-Expert 7.0 (Trial version, State-Ease Inc., Minneapolis, MN, USA) software was used to obtain the coefficients of the quadratic polynomial model. The quality of the fitted model was expressed by the coefficient (*R*^2^) of determination, and its statistical significance was checked by *F*-test. All the analyses were performed in triplicate. Data are expressed as mean ± standard deviation (*n* = 3).

## 4. Conclusions

Box–Behnken response surface design coupled with numerical optimization analysis results showed the effects of the variables (extraction temperature and number of cycles) were significant and quadratic models were obtained for predicting responses. The optimum conditions determined were as follows: extraction time of 6.5 min, extraction temperature of 71 °C, and number of cycles of 3. Under these optimal conditions, a maximum crude polysaccharide yield of 19.77 ± 0.12% can be achieved, which accorded well with the value that was predicted by the model. The ASE-RSM new industrial technologies proved to be promising candidates in the valorization of chestnut mushroom by-product. In addition, chestnut polysaccharide should be a potential antioxidant for the development of natural antioxidants.

## Figures and Tables

**Figure 1 molecules-27-02380-f001:**
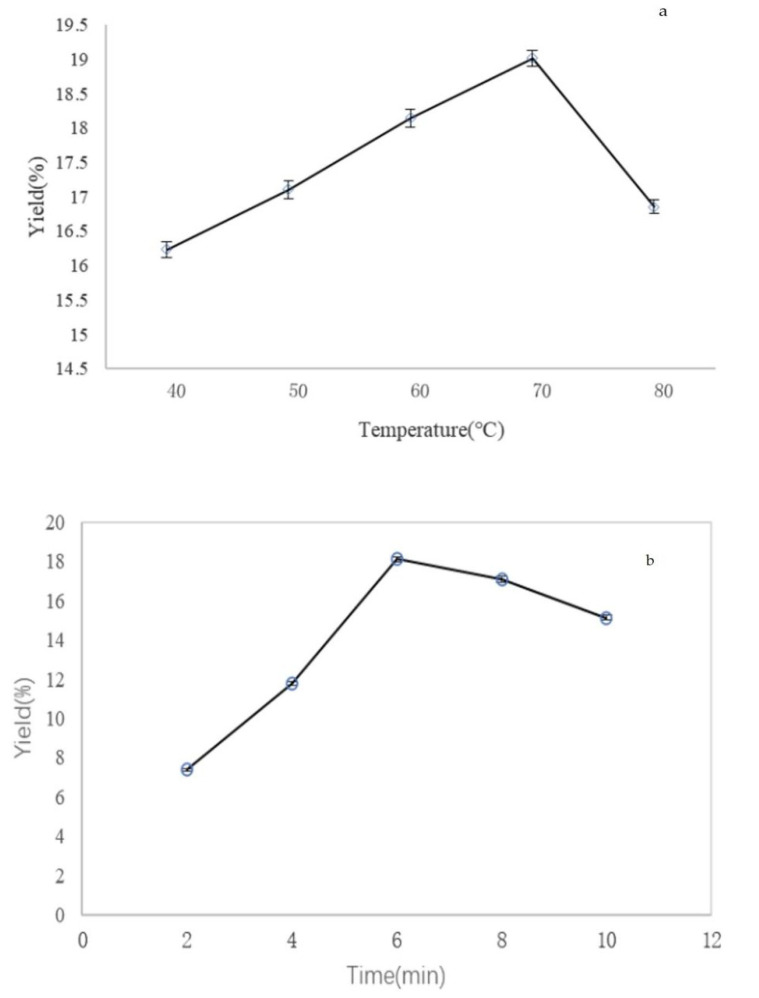
Effects of different extraction parameters on extraction yield of crude polysaccharide from mushroom (**a**) extraction temperature, °C; (**b**) extraction time, min; (**c**) number of cycles.

**Figure 2 molecules-27-02380-f002:**
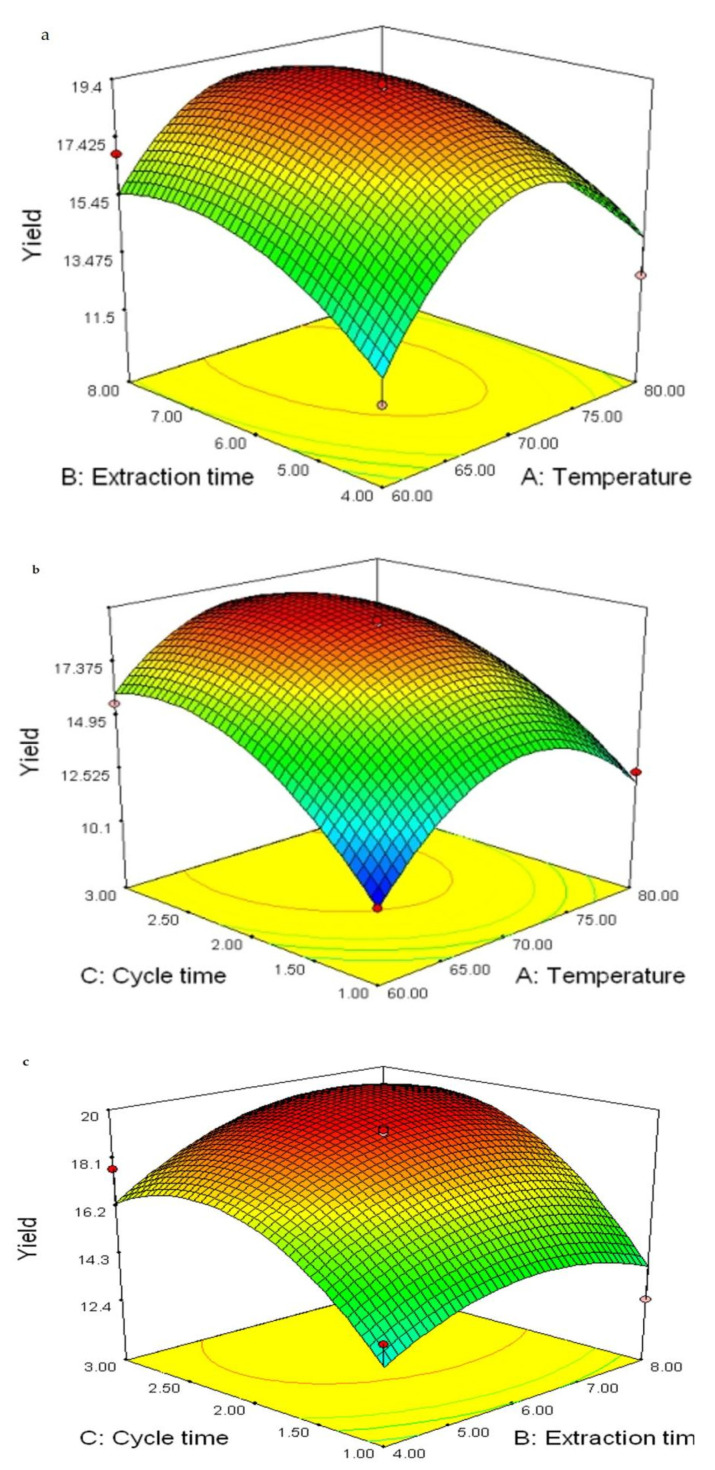
Response surface plots showing the interaction effects of extraction time and extraction temperature (**a**), number of cycles and extraction temperature (**b**), and number of cycles and extraction time (**c**) on the yield of crude polysaccharide.

**Figure 3 molecules-27-02380-f003:**
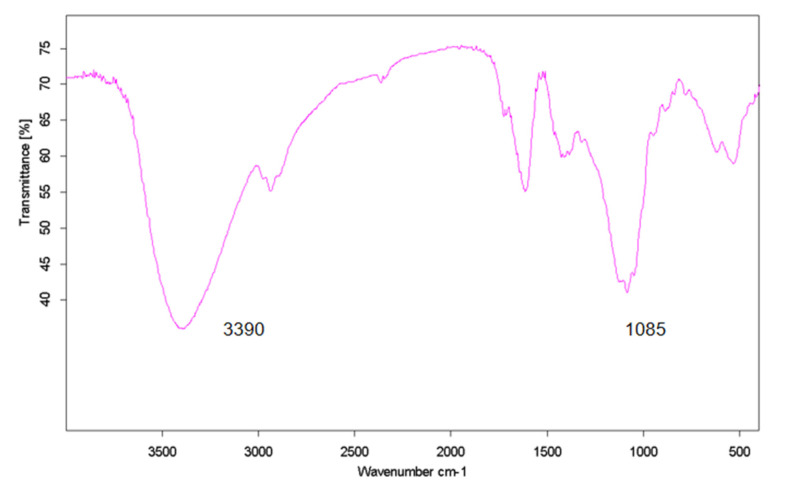
FT-IR spectrum of crude polysaccharide in the range of 4000–400 cm^−1^.

**Figure 4 molecules-27-02380-f004:**
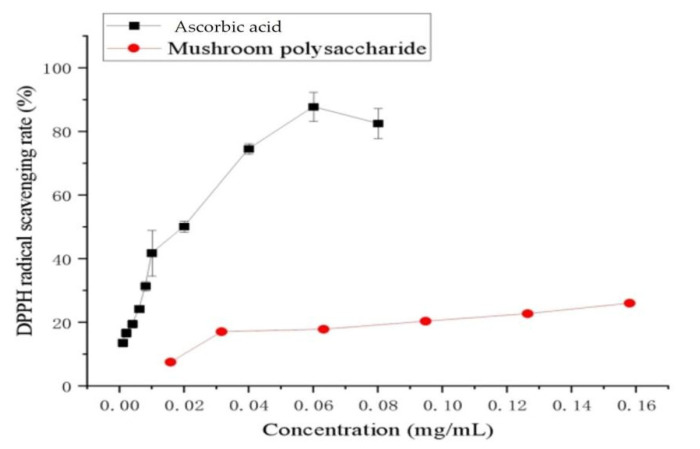
In vitro antioxidant activity assessment DPPH radical scavenging activity of polysaccharide from chestnut mushroom. Values are expressed as mean ± SD (*n* = 3).

**Figure 5 molecules-27-02380-f005:**
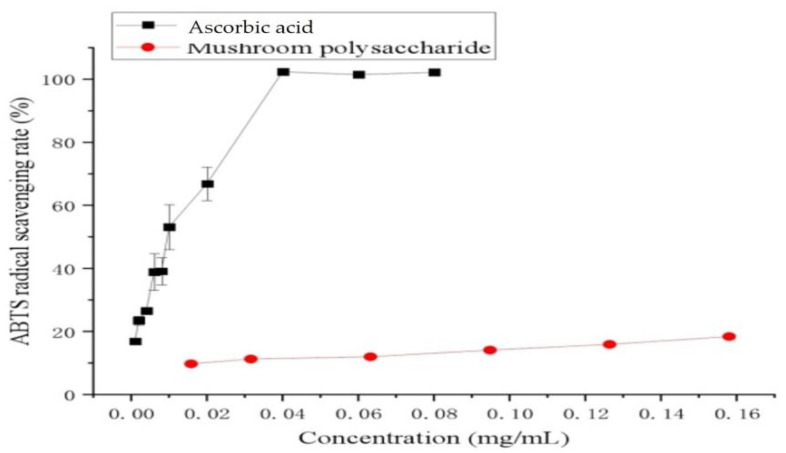
In vitro antioxidant activity assessment of ABTS radical scavenging activity of polysaccharide from chestnut mushroom. Values are expressed as mean ± SD (*n* = 3).

**Table 1 molecules-27-02380-t001:** Yields of crude polysaccharide with different extraction methods.

Extraction Methods	Extraction Time	Extraction Temperatures	Extraction Pressure (MPa)	Extraction Power (W)	Yield of Crude Polysaccharide (%)
ASE	6.5 min	71 °C	10	-	19%
CRE	2 h	100 °C	-	-	3.2%
UAE	1 h	50 °C	-	100	1.3%

**Table 2 molecules-27-02380-t002:** Box–Behnken design and observed responses ^a^.

Run	Independent Variable			Response (*Y*%)
	*A* (Temperature, °C)	*B* (Time, min)	*C* (Number of Cycles)	
1	60 (−1)	4 (−1)	2 (0)	11.56 ± 0.02
2	80 (+1)	4 (−1)	2 (0)	12.72 ± 0.04
3	60 (−1)	8 (+1)	2 (0)	16.92 ± 0.05
4	80 (+1)	8 (+1)	2 (0)	15.33 ± 0.09
5	60 (−1)	6 (0)	1 (−1)	10.22 ± 0.09
6	80 (+1)	6 (0)	1 (−1)	12.38 ± 0.03
7	60 (−1)	6 (0)	3 (+1)	15.52 ± 0.05
8	80 (+1)	6 (0)	3 (+1)	14.74 ± 0.04
9	70 (0)	4 (−1)	1 (−1)	13.75 ± 0.02
10	70 (0)	8 (+1)	1 (−1)	12.44 ± 0.01
11	70 (0)	4 (−1)	2 (0)	17.72 ± 0.08
12	70 (0)	8 (+1)	3 (+1)	18.20 ± 0.07
13	70 (0)	6 (0)	2 (0)	19.22 ± 0.05
14	70 (0)	6 (0)	2 (0)	19.29 ± 0.03
15	70 (0)	6 (0)	2 (0)	19.23 ± 0.03
16	70 (0)	6 (0)	2 (0)	19.20 ± 0.04
17	70 (0)	6 (0)	2 (0)	19.10 ± 0.09

^a^: Average value of triplicate experiments.

**Table 3 molecules-27-02380-t003:** ANOVA for the fitted quadratic polynomial model of extraction of crude polysaccharide.

Source	Sum of Squares	DF	Mean Square	*F*-Value	*p*-Value Prob > *F*
Model	145.26	9	16.14	10.64	0.0025
Residual	10.61	7	1.52		
Lack of fit	10.60	3	3.53	784.11	<0.0001
Pure error	0.018	4	4.50 × 10^−3^		
Cor total	155.87	16			
	*R*^2^ = 0.9319		CV = 7.82		

**Table 4 molecules-27-02380-t004:** Estimated regression model of relationship between response variable (yield of crude polysaccharide) and independent variables (*A*, *B*, and *C*).

Variables	Sum of Squares	DF	Mean Square	*F*-Value	*p*-Value Prob > *F*
*A*	0.12	1	0.12	0.076	0.7906
*B*	6.35	1	6.35	4.19	0.0799
*C*	37.79	1	37.79	24.92	0.0016
*AB*	1.89	1	1.89	1.24	0.3.13
*AC*	2.16	1	2.16	1.43	0.2713
*BC*	0.79	1	0.79	0.52	0.4928
*A* ^2^	57.46	1	57.46	37.89	0.0005
*B* ^2^	8.05	1	8.05	5.31	0.0547
*C* ^2^	22.24	1	22.24	14.67	0.0065

## Data Availability

Not applicable.

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
