# Peer review of "Optimization Extraction and Antioxidant Activity of Crude Polysaccharide from Chestnut Mushroom (Agrocybe aegerita) by Accelerated Solvent Extraction Combined with Response Surface Methodology (ASE-RSM)"

_molecules, 2022, doi:10.3390/molecules27082380_

Round 1
Reviewer 1 Report
This paper describes the extraction and antioxidant activity of polysaccharide from chestnut mushroom (Agrocybe aegerita) by accelerated solvent extraction combined with response surface methodology (ASE-RSM). Some specific comments:
- p.3, line 100: The maximum yield was reached at 70°C, so why was the temperature fixed at 60°C for the preliminary tests? I suggest including fixed conditions in the caption of Figure 1.
- p.3, line 105-111: What would be the reason for the decrease in the extraction yield with more than two cycles? The number of cycles was a significant factor affecting extraction yield, so consider better explaining this effect.
- p.4, line 121: Please revise the sentence “Table 2 shows that the process variables and experimental data.”
- Figure 3: please comment about bands in the range 1700-1500 cm-1. The bands around this region and bands around 3000 cm-1 help characterize polysaccharide functional groups. The y-axis scale is usually reported as arbitrary units, eg. Transmittance (A.U.).
- Figure 4 and 5: the legends must be improved.
- p.8, line 229: “Moreover, the different results might be related with the different nature of dietary fibre (mushroom and wheat)”. It is not clear why this comparison was mentioned here.
- p.12, line 352: Temperature also showed a significant quadratic effect.
Author Response
Manuscript ID: molecules-1625165  
Type of manuscript: Research Paper
Title : Optimization extraction and antioxidant activity of crude polysaccharide from chestnut mushroom (Agrocybe aegerita) by accelerated solvent extraction combined with response surface methodology (ASE-RSM)
Dear Editors and Reviewers: We would like to thank you for your time and effort in reviewing our manuscript, and for your suggestion in improving the quality of our manuscript. We rephrased Result and discussion section. We revised this manuscript according to the editors’ and reviewers’ comments. The following is a point-by-point response to the reviewers’ comments. All revised items were in “Track Changes” in the new version according to reviewer’s suggestions. In addition, authors also ask for one native speaker to revise this manuscript.
Response to comments from reviewer #1
Comment: This paper describes the extraction and antioxidant activity of polysaccharide from chestnut mushroom (Agrocybe aegerita) by accelerated solvent extraction combined with response surface methodology (ASE-RSM). Some specific comments:
- p.3, line 100: The maximum yield was reached at 70°C, so why was the temperature fixed at 60°C for the preliminary tests? I suggest including fixed conditions in the caption of Figure 1.
Response: We must thank you and the reviewers for the critical comments and feedback. We finished the effect of extraction temperature on yield of crude polysaccharide and the maximum yield was reached at 70°C, so we fixed at 70°C for the preliminary tests and RSM tests. We apologize for this error.
Comment: - p.3, line 105-111: What would be the reason for the decrease in the extraction yield with more than two cycles? The number of cycles was a significant factor affecting extraction yield, so consider better explaining this effect.
Response: We must thank you very much for the kind comments. We explained the more number of cycles had less yield of crude polysaccharide in text.
Comment: - p.4, line 121: Please revise the sentence “Table 2 shows that the process variables and experimental data.”
Response: Thank you very much for your kind comments. We changed “Table 2 shows that the process variables and experimental data.” to “Table 2 shows the process variables and experimental data.”
Comment: - Figure 3: please comment about bands in the range 1700-1500 cm-1. The bands around this region and bands around 3000 cm-1 help characterize polysaccharide functional groups. The y-axis scale is usually reported as arbitrary units, eg. Transmittance (A.U.).
Response: We must thank you very much for the critical comments. We added the range 1700-1500 cm-1 for signal.
Comment: - Figure 4 and 5: the legends must be improved.
Response: Thank you very much for your kind comments. We tried our best to improve the quality of Figure 4 and 5.
Comment: - p.8, line 229: “Moreover, the different results might be related with the different nature of dietary fibre (mushroom and wheat)”. It is not clear why this comparison was mentioned here.
Response: We appreciate the important comments. We revised this sentence to help readers understand easily.
Comment: - p.12, line 352: Temperature also showed a significant quadratic effect.
Response: We appreciate for your valuable suggestion. We added temperature in this sentence.
Reviewer 2 Report
Initially, the extraction yields, to crude polysaccharide, are compared using the conventional reflux extraction, the ultrasound-assisted extraction, and the accelerated solvent extraction. The experimental data showed that the third technique was the most effective. Using the response surface methodology, the optimal conditions were proposed to maximize the extraction yield. Finally, the antioxidant activity of the isolated polysaccharide, using the acceleration solvent extraction technique, was determined using the DPPH and ABTS assays.The work is very well structured and understandable.I have two remarks to make.
- Line 254-255. What does “The samples were dried under food grade conditions” mean? I do not think there is such a scientific term. Please provide drying conditions (technique, temperature, time).
- In Figure 3 it is advisable to show three representative spectra of the isolated crude polysaccharide with the three extraction techniques and to discuss the possible differences.
Author Response
Manuscript ID: molecules-1625165  
Type of manuscript: Research Paper
Title : Optimization extraction and antioxidant activity of crude polysaccharide from chestnut mushroom (Agrocybe aegerita) by accelerated solvent extraction combined with response surface methodology (ASE-RSM)
Dear Editors and Reviewers: We would like to thank you for your time and effort in reviewing our manuscript, and for your suggestion in improving the quality of our manuscript. We rephrased Result and discussion section. We revised this manuscript according to the editors’ and reviewers’ comments. The following is a point-by-point response to the reviewers’ comments. All revised items were in “Track Changes” in the new version according to reviewer’s suggestions. In addition, authors also ask for one native speaker to revise this manuscript.
Response to comments from reviewer #2
Comment: Initially, the extraction yields, to crude polysaccharide, are compared using the conventional reflux extraction, the ultrasound-assisted extraction, and the accelerated solvent extraction. The experimental data showed that the third technique was the most effective. Using the response surface methodology, the optimal conditions were proposed to maximize the extraction yield. Finally, the antioxidant activity of the isolated polysaccharide, using the acceleration solvent extraction technique, was determined using the DPPH and ABTS assays.The work is very well structured and understandable.I have two remarks to make.
-Line 254-255. What does “The samples were dried under food grade conditions” mean? I do not think there is such a scientific term. Please provide drying conditions (technique, temperature, time).
Response: We thank the reviewer for this suggestion. We added the scientific term for drying technique, temperature and time in text.
Comment: -In Figure 3 it is advisable to show three representative spectra of the isolated crude polysaccharide with the three extraction techniques and to discuss the possible differences.
Response: We thank the reviewer for raising the important point, and the primary structure of crude polysaccharide was characterized using FT-IR. The primary structure of crude polysaccharide was consistent using different extraction methods.
Reviewer 3 Report
The manuscript entitled “Title Optimization extraction and antioxidant activity of crude polysaccharide from chestnut mushroom (Agrocybe aegerita) by accelerated solvent extraction combined with response surface methodology (ASE-RSM)” reports the optimization of ASE of crude polysaccharide from chestnut mushroom by response surface methodology and its comparison with conventional and ultrasound-assisted extraction. The manuscript is interesting, however, it needs to be revised prior to publication.
In the abstract, the authors do not mention that they also tested other extraction methods such as Conventional reflux extraction (CRE) Ultrasound-assisted extraction (UAE). Please revise.
Results and Discussion section:
In Table 1, the authors reported the ASE optimized values, therefore item “2.1. Comparison of different extraction methods” should be placed after the optimization studies.
Why did the authors use 2 h for CRE and 1 h for UAE? Please explain.
Could the lower temperature of the UAE be responsible for the lower yields, since the temperature is one of the process parameters that affect the extraction yields? Please explain.
Please add the statistical significant letter in Figure 1.
Why did the authors show equation 2 with all the coefficients since some of them are not statistically significant? Please explain
The model has a significant lack of fit. The F calculate of the lack of fit divided by the pure error (784) is higher than the F tabulated for 3,4 at 0.05% that is 4.19, meaning that the Flack of fit/pure error > Ftab, therefore there is a lack of fit. Please revise your statement. The lack of fit arises from the low variability in the central points, low pure error. But the authors should rewrite their statement because it is not in accordance with their results.
Line 78: Please, add a space in “ofcrude”
Please add some chemical explanation for the results from the optimization, why do higher temperatures favor extraction? How the number of cycles can affect the system? Etc
Figure 4 legend is not clear. Please revise it.
How did the authors select the process parameters to be optimized? What about the solids to liquid ratio? Extraction pH? Sample particle size? Solids to liquid ratio is known to significantly affect extraction yields. Please explain.
Material and methods section:
Were the CSE and the UAE performed both with 10 mg of sample plus 150 mL of water? What about the ASE? Please improve clarity in the text.
Did the authors use the extract with no dilution for the DPPH and ABST assays? Please add more information.
Please add the results of ABTS and DPPH for the other extraction methods tested.
Author Response
Manuscript ID: molecules-1625165  
Type of manuscript: Research Paper
Title : Optimization extraction and antioxidant activity of crude polysaccharide from chestnut mushroom (Agrocybe aegerita) by accelerated solvent extraction combined with response surface methodology (ASE-RSM)
Dear Editors and Reviewers: We would like to thank you for your time and effort in reviewing our manuscript, and for your suggestion in improving the quality of our manuscript. We rephrased Result and discussion section. We revised this manuscript according to the editors’ and reviewers’ comments. The following is a point-by-point response to the reviewers’ comments. All revised items were in “Track Changes” in the new version according to reviewer’s suggestions. In addition, authors also ask for one native speaker to revise this manuscript.
Response to comments from reviewer #3
Comment: The manuscript entitled “Title Optimization extraction and antioxidant activity of crude polysaccharide from chestnut mushroom (Agrocybe aegerita) by accelerated solvent extraction combined with response surface methodology (ASE-RSM)” reports the optimization of ASE of crude polysaccharide from chestnut mushroom by response surface methodology and its comparison with conventional and ultrasound-assisted extraction. The manuscript is interesting, however, it needs to be revised prior to publication.
In the abstract, the authors do not mention that they also tested other extraction methods such as Conventional reflux extraction (CRE) Ultrasound-assisted extraction (UAE). Please revise.
Response: We are very sorry for this lack of information. We rewrote the abstract section according to the reviewer’s comments.
Comment: Results and Discussion section:
In Table 1, the authors reported the ASE optimized values, therefore item “2.1. Comparison of different extraction methods” should be placed after the optimization studies.
Response: Thanks for your kind suggestion very much. We compared the different extraction methods and obtained the ASE method with highest yield. So we optimized this method using RSM.
Comment: Why did the authors use 2 h for CRE and 1 h for UAE? Please explain.
Response: We thank the reviewer for pointing this out. The conventional reflux extraction with short time could not had the high yield. So we use 2 hours for conventional reflux extraction and 1 hour for UAE.
Comment: Could the lower temperature of the UAE be responsible for the lower yields, since the temperature is one of the process parameters that affect the extraction yields? Please explain.
Response: We thank the reviewer for this question. In this manuscript, we performed UAE for 1 hour and ASE only for 6.5 min. The temperature plays a important role in crude polysaccharide extraction. The temperature rose in 1 hour using UAE method.
Comment: Please add the statistical significant letter in Figure 1.
Response: We thank the reviewer for pointing this out. We provided error bars (mean ± SD) in Figure 1.
Comment: Why did the authors show equation 2 with all the coefficients since some of them are not statistically significant? Please explain
Response: We appreciate for your valuable suggestion. We got the equation 2 from the software Design-Expert 7.0. It is a second-order polynomia equation for all the coefficients in RSM.
Comment: The model has a significant lack of fit. The F calculate of the lack of fit divided by the pure error (784) is higher than the F tabulated for 3,4 at 0.05% that is 4.19, meaning that the F lack of fit/pure error > F tab, therefore there is a lack of fit. Please revise your statement. The lack of fit arises from the low variability in the central points, low pure error. But the authors should rewrite their statement because it is not in accordance with their results.
Response: We thank the reviewer for the constructive comments that surely helped in improving our manuscript. We rewrote this section for improving this manuscript.
Comment: Line 78: Please, add a space in “ofcrude”.
Response: We are sorry for this type error. We revised “ofcrude”to “of crude”.
Comment: Please add some chemical explanation for the results from the optimization, why do higher temperatures favor extraction? How the number of cycles can affect the system? Etc
Response: We thank the reviewer for this question. We added these contents in Result and Discussion section.
Comment: Figure 4 legend is not clear. Please revise it.
Response: Thank you very much for your kind comments. We tried our best to improve the quality of Figure 4.
Comment: How did the authors select the process parameters to be optimized? What about the solids to liquid ratio? Extraction pH? Sample particle size? Solids to liquid ratio is known to significantly affect extraction yields. Please explain.
Response: We thank the reviewer for pointing this out. For optimization of extraction technology study, there are many parameters including extration temperature, extraction time, solids to liquid ratio, number of cycles ect. In this manuscript, we selected the different parameters through several factors. For ASE method, the number of cycles is more vital than other parameters. In addition, we use the consistent particle size (through 40-mesh sieve). And we use water (about pH 7.0) as solvent systems, it is consumed environmentally friendly for application.
Comment: Material and methods section:
Were the CSE and the UAE performed both with 10 mg of sample plus 150 mL of water? What about the ASE? Please improve clarity in the text.
Response: We appreciate for your valuable suggestion. The CSE and UAE were performed with 10 mg of sample plus 150 mL of water. For ASE, we used 10 mg of sample plus 100 mL of water.
Comment: Did the authors use the extract with no dilution for the DPPH and ABST assays? Please add more information.
Response: We thank the reviewer for the constructive comments that surely helped in improving our manuscript. For DPPH and ABTS assays, we made 1 mg mL-1 solution using the crude polysaccaride.
Comment: Please add the results of ABTS and DPPH for the other extraction methods tested.
Response: We appreciate the reviewer for this question. We carried out the DPPH and ABTS assays using crude polysaccharide from optimized ASE method.
Round 2
Reviewer 1 Report
Most comments were addressed. I have one minor comment:
p. 3, lines 112-114: Please rewrite. The sentence is repetitive and it is not clear.
Reviewer 3 Report
The authors significantly improved the quality of their manuscript and replied to all the questions raised by this reviewer. Therefore, I recommend its publication.